# Deep-Learning Forecasting Method for Electric Power Load via Attention-Based Encoder-Decoder with Bayesian Optimization

**Xue-Bo Jin** [1,2] , **Wei-Zhen Zheng** [1] , **Jian-Lei Kong** [1,2,*] , **Xiao-Yi Wang** [1,2] , **Yu-Ting Bai** [1] , **Ting-Li Su** [1] **and Seng Lin** [3,*]

1 School of Artificial Intelligence, Beijing Technology and Business University, Beijing 100048, China; jinxuebo@btbu.edu.cn (X.-B.J.); zhengweizhen@st.btbu.edu.cn (W.-Z.Z.); wangxy@btbu.edu.cn (X.-Y.W.); baiyuting@btbu.edu.cn (Y.-T.B.); sutingli@btbu.edu.cn (T.-L.S.)

2 National Key Laboratory of Environmental Protection Food Chain Pollution Prevention, Beijing 100048, China

3 Beijing Research Center of Intelligent Equipment for Agriculture, Beijing 100097, China

* Correspondence: kongjianlei@btbu.edu.cn (J.-L.K.); linseng@nercita.org.cn (S.L.)

**Abstract:** Short-term electrical load forecasting plays an important role in the safety, stability, and sustainability of the power production and scheduling process. An accurate prediction of power load can provide a reliable decision for power system management. To solve the limitation of the existing load forecasting methods in dealing with time-series data, causing the poor stability and non-ideal forecasting accuracy, this paper proposed an attention-based encoder-decoder network with Bayesian optimization to do the accurate short-term power load forecasting. Proposed model is based on an encoder-decoder architecture with a gated recurrent units (GRU) recurrent neural network with high robustness on time-series data modeling. The temporal attention layer focuses on the key features of input data that play a vital role in promoting the prediction accuracy for load forecasting. Finally, the Bayesian optimization method is used to confirm the model's hyperparameters to achieve optimal predictions. The verification experiments of 24 h load forecasting with real power load data from American Electric Power (AEP) show that the proposed model outperforms other models in terms of prediction accuracy and algorithm stability, providing an effective approach for migrating time-serial power load prediction by deep-learning technology.

**Keywords:** electric power load prediction; deep-learning encoder-decoder framework; gated recurrent neural units; temporal attention; Bayesian optimization

## 1. Introduction

With the rapid economic and social development, electric power plays an increasingly important role in all aspects of humanity's domestic life and industrial and commercial practical applications. Especially in recent decades, various information and intelligent products have gone deep into everybody's daily life with further expanding the demand for lots of electricity. However, the electric-power industry and commercial companies have struggled to provide high-quality, fair-priced, stable, and safe power supply to mass end-consumers, since electrical energy is an instant energy source, which is hardly stored in large quantities for a long time [1]. Additionally, some special issues such as weather changes, holidays, and other unexpected events have changed the electricity consumption patterns with electricity demand soaring [2]. For example, in the context of the COVID-19 pandemic spreading across the globe, China has become the only normal economy and the major production base for the global fight against the epidemic due to the effective government governance and the joint efforts of the whole nation. A large number of a production order for epidemic vaccines and prevention materials, coupled with drastic cooling in cold weather this winter, have caused insufficient electric power supply in partial areas of Chinese southern, even though China is the country with the largest electricity production and the largest increase in the world [3]. To deal with these constant changes

between electricity generation and consumption, it is expected that implementing efficient management operations in the electronic power systems to balance the electricity supply and demand as much as possible. Therefore, constructing accurate, robust, and fast models forecasting electric power load became the fundamental approach to achieve reliable and high-efficiency operational management of abundant power utilities, such as electricity production planning, high-voltage transmission decision-making, power load dispatch, and so on [4,5].

According to the length of forecast time, the power load forecasting models can be categorized as short-term, medium-term, and long-term in terms of their predictive performance. The short-term load forecasting (STLF) models predict electricity changes within a week or even one day for scheduling the production and delivery. Medium-term forecasting models are designed to predict electricity usage from 1 week to 1 year. Long-term ones mean power prediction for more than one year, applied to plan fuel consumption or develop annual power supply on the macroscopic perspective. The short-term forecasting models are most relevant to electric load prediction with the considerable significance of planning efficient operations and reducing power waste, which also provides a reliable decision basis for economic management and sustainable development of the whole power system [6].

However, the STLF is a very complex nonlinear temporal issue affected by various internal and external factors, making it challenging to accurately predict the electricity load's variation trend of electricity load. Thereby, it is extremely necessary to utilize state-of-the-art mathematical models to efficiently assist in uplifting the prediction performance of intelligent management systems and prevent potential risks in the electrical industry. Prediction algorithms for acquiring knowledge of future trends in electricity load changes are statistical parameter estimation methods, shallow-structured representative models, and deep learning models. In statistical parameter estimation methods, the mathematical correlation and physical information of complex buildings are statistically figured out to implement the input-to-output mapping based on nonlinear, tremulous, and periodic characteristics. The establishment of statistical models includes multiple regression analysis [7], Kalman filtering [8], exponential smoothing [9], weighted moving average [10], auto-regressive (AR) [11], autoregressive moving average (ARMA) [12], autoregressive integrated moving average (ARIMA) [13], and so on. These models usually have a complete theoretical derivation process and modeling steps, which require a priori knowledge for empirical assumptions to mining data and determine the parameters [14], which are often difficult to predict the outcome of power load when dealing with complex nonlinear data or mismatching of data distribution and model Hypothesis.

In contrast, shallow-structured representative models with parametric self-learning and non-linear adaptive are usually more suitable for complex time serial issues, which have been widely used in the field of power load forecasting [15]. Those models mainly consist of various machine-learning methods with classical shallow structures, including k-nearest neighbors (k-NN) [16], decision trees (DT) [17], support vector machines (SVM) [18], artificial neural networks (ANNs) [19], fuzzy set, etc. Furthermore, more research tries to integrate different subsidiary models to form an effective hybrid model. With preserving each model's advantages, this method is shown to have a good performance in obtaining predetermined rules from complex historical data and improving the prediction effect. For example, Fan G proposes a support vector regression (SVR) model with the AR method for power load prediction [20]. And Pal, S used a generalized fuzzy set to adjust the hyper-parametric weight of the backpropagation (BP) neural network for the next 24 h prediction of the power load [21]. Similarly, Wang proposed a short-term load prediction method based on an improved ANNs with the decomposed learning way [22]. The results have been shown that the prediction accuracy can be improved by decomposing into multiple components and modeling separately with predictors. Machine-learning-based methods gain a limited performance for non-linear load sequences but still have some drawbacks. All of them require handcrafted features with too much human intervention, making it

difficult to appropriately capture vital non-linear relationships and underlying temporal dependencies between the outputs and inputs based on the available load training data.

In recent years, with the global shift of electric supply networks rapidly towards Industry 4.0 and mart IoT management, the deep-learning methods are playing a vital role in this transition. The core concept of deep-learning technology is stacking multi-layered neural network with massive high-quality annotated data sets (such as ImageNet, MSCOCO, etc.) and various training tricks (such as Mixup data enhance, Residual structure, Leaky-ReLU function, Transformer, etc.), which have been demonstrated to be effective for handling non-linear, dynamic, and complicated problems in all aspects of living life [23]. In many areas, deep-learning methods have made remarkable progress in image processing, video tracking, speech recognition, and natural language understanding [24–27]. Meanwhile, some researchers have pointed out that deep-learning methods including Deep Belief Networks (DBNs), Convolutional Neural Networks (CNNs), Recurrent Neural Networks (RNNs), Generative Adversarial Networks (GAN), Graph Neural Network (GNN), etc., are very valuable to carry out better power load prediction by making full use of massive time-series data [28–30]. In particular, the powerful type of deep-learning framework, RNNs, specially designed for temporal analysis and modeling, have already gained a great amount of concern due to their flexibility in obtaining underlying non-linear relationships and sequential rules [31,32]. In recent years, RNNs have widely shown their STLF applications' success with their unique structure [33,34]. However, traditional RNNs suffer from vanishing gradients, which make them easily lost in local extreme values and lack the competence of capturing long-term dependencies [35]. To improve the time-serial predicting accuracy of traditional RNNs, long short-term memory units (LSTM) have overcome those limitations by import a computing mechanism of input, forgetting, and output gates and achieved great success in various electricity load applications [36].

For instance, an LSTM-based forecasting model was developed to predict the short-term electricity consumption of individual residential customers [37]. Similarly, Xiangyun constructed the prediction model by using the LSTM network to predict the day-ahead electricity change for solar energy supply in the region of Santiago Island. Some studies have extended the work on electricity forecasting to probabilistic forecasting, taking full account of the uncertainty of forecasting and transferring the value of forecasting accuracy to estimating the probability of each future possibility [38,39]. Recently, another variant of RNNs, the gated recurrent units (GRU), is also gradually applied in the field of power sequence predicting. With a simpler neuron structure than LSTM by fusing forgetting and input gates into a single update gate, GRU makes its calculation speed faster with better expression ability for sequence electric load data. Wang proposes a novel approach to predict short-term photovoltaic power based on a gated recurrent unit, which effectively considers the influence of historical features on the future output [40]. Afrasiabi designed an end-to-end composite model consisting of GRU and convolutional neural network (CNN) for probabilistic residential load prediction [41].

Although RNN-based methods have achieved substantial power load forecasting, it is natural to consider other state-of-the-art networks such as encoder-decoder networks for time series prediction. Based upon several RNN layer units, the encoder-decoder methods have become the popular sequence-to-sequence (seq2seq) architecture [42] due to their success in the fields of natural language processing, machine translation, and so on. Encoder-decoder usually contents an encoder and a decoder consisting of multiple RNN layers to encode the source data as a fixed-length vector. Then, it uses the decoder to generate a translation, which effectively extracts the time-series characteristics and transformation features from the input data. Existing studies have proven that using the encoder-decoder structures achieves better performance on real-time prediction issues. Malhotra has developed a multi-sensor anomaly detector based on an encoder-decoder model using LSTM as the codec component [43]. Qin uses a dual-stage encoder-decoder model to effectively analyze the implicit pattern of predicting ambient temperature and stock prices [44]. In recent years, many studies have begun to shift their attention to

applying the Encoder-decoder models to solve power forecasting and related management problems. Bottieau used the seq2seq models to make probabilistic predictions about the single imbalance pricing in European electricity markets [45]. Mashlakov using multi-attention recurrent neural network for multi-step forecasting of battery state-of-charge [46]. Sehovac used the GRU-based seq2seq model to predict the short, medium, and long-term power data with better forecasting performance [47]. Some researchers have also applied an optimized encoder-decoder network with temporal attention to adaptively learn long-term dependency and hidden correlation features for handling multivariate temporal forecasting problems [48].

Although exsiting encoder-decoder models and other seq2seq networks have shown their efficacy for application in various fields, they may not be suitable for massive and time-serial power data. Two main drawbacks are restricting the expanded application of encoder-decoder in power management. One problem with encoder-decoder methods is that their performance will be seriously influenced by hyperparameters based on experiential knowledge or many attempts to select parameters for better results. This operation spends a lot of experimenting time and computing resources, and the hyperparameters obtained may not be optimal, resulting in the instability and limited accuracy of models [49]. Therefore, designing an effective strategy to find the optimal model hyperparameters is an essential step to solve the above issue, which is why we motivate us to investigate the Bayesian optimization algorithm in this paper.

Another problem is that encoder-decoder models' performance will deteriorate rapidly as the length and quantity of input sequence increases. When non-linear electric time series consists of multiple internal variables and exogenous impact factors, the encoder-decoder networks cannot explicitly select relevant key information to make accurate predictions under different conditions. To resolve this issue, this paper considers effective encoder-decoder modeling from the perspective of attention mechanism, which selects parts of hidden states across all the time steps to fit electric time series analysis and forecasting.

In response to these issues above, and inspired by attention mechanism and parameter optimization strategy, an attention-based codec prediction model with Bayesian optimization is proposed for the first time, which aims to achieve the purpose of adaptively learning the implicit temporal dependency features and improving prediction performance of complexity power load forecasting. With contrastive validation on the American Electric Power dataset, this proposed model has good prediction results with outstanding robustness. Major contributions of this study are shown below:

(1) Improve the prediction model's overall performance for electrical load by designing appropriate GRU-based encoder-decoder architecture incorporating temporal attention mechanism, adjusting the non-linear degree and dynamic adaptability of the network.

(2) Replace the previous manually selected ways, the Bayesian optimization algorithm is utilized to automatically assure the hyperparameters of encoder-decoder model, which results in improving prediction performance and training efficiency of seq2seq method with too many parameters.

The rest is organized as follows. Section 2 outlines material and methods to illustrate the internal structure of the proposed prediction method. Section 3 describes the settings of models, forecasting results, and comparative analysis. Finally, we conclude the paper with the possible future implications in Section 4.

## 2. Materials and Methods

### 2.1. Traditional Encoder-Decoder Structure

The encoder-decoder prediction model consists of two parts, encoder, and de-encoder, and consists of multiple layers of arbitrary types of neural units, such as MLP, CNN, RNN, LSTM, GRU, etc. [50]. In this paper, we use GRU neural cell to construct an encoder-decoder prediction model. In the training phase, GRUs in the encoder is used to convert the input power load data as an encoding vector. Then, the GRUs-based decoder transfers

the encoded vector to obtain the forecasting values in the prediction phase. The traditional encoder-decoder framework is shown in Figure 1.

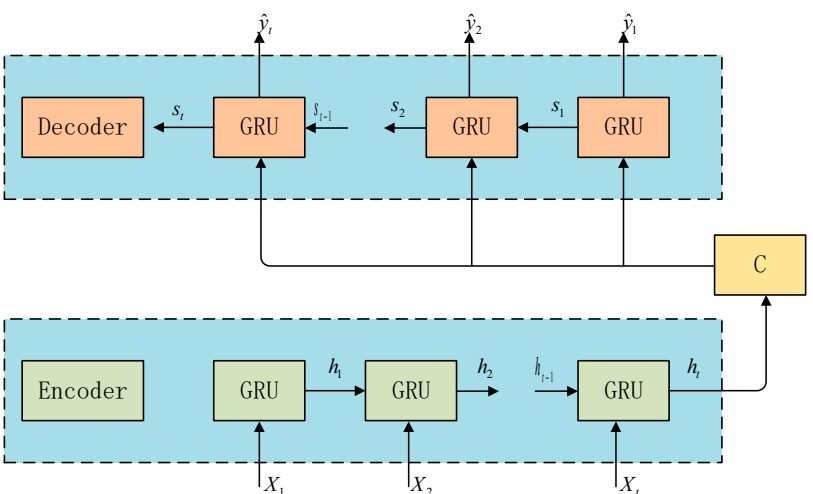

**Figure 1.** Traditional encoder-decoder prediction structure.

Assume that $X = [X_1, X_2, \cdots, X_t]^T$ is the input power load sequence, $h = [h_1, h_2, \cdots, h_t]^T$ and $s = [s_1, s_2, \cdots, s_t]^T$ represents the hidden layer states in the encoder and decoder sections respectively. In the encoder stage, the hidden state $h_{t-1}$ of the previous neuron unit is calculated with the current input $X_t$ to derive the next hidden state $h_t$:

$$h_t = f(h_{t-1}, X_t) \tag{1}$$

where $f(\cdot)$ denotes the recurrent computation of neural network units. The hidden state $h_t$ of the last time step is followed by the output encoding vector $C$, which is then fed into the decoder as the input to the decoder for the codec. In the decoder stage, the GRU network takes the encoding vector $C$ as input and computes the predicted values.

$$\hat{y}_t = g(s_{t-1}, C) \tag{2}$$

where $g(\cdot)$ denotes the activating operation of the RNN unit. $\hat{y}_t$ indicates the prediction results, which will be used to compare with real labels to objectively evaluate the accuracy performance of the entire model.

### 2.2. Temporal Attention Layer

Though the traditional encoder-decoder framework is very classical and widely used. However, it is limited in its information represented by the loss of information caused by fixed-length encoding. The middle encoding vector is mounted in vector dimensions for fixation regardless of the variational length of the input or output sequence, which cannot fully represent the overall information for longer input sequences. This results in the loss of key features and leads to gradient degradation of the neural network [51,52]. In contrast to the classical encoder-decoder, this paper introduces the temporal attention layer to represent the connecting proposes between each step output of the encoder and each generation step of the decoder. This attention-based layer is implemented by two parts: the output of the encoder, and the hidden state of the decoder. The hidden states of each step in the encoder are output, which is incorporated with the hidden state of each step in the decoder for calculating the attention weights. The encoder-decoder model based on the attention mechanism removes the fixed-length encoding bottleneck, and the information is passed from the encoder to the decoder without losing any key information. The structure of the temporal attention layer added to the encoder-decoder model is shown in Figure 2.

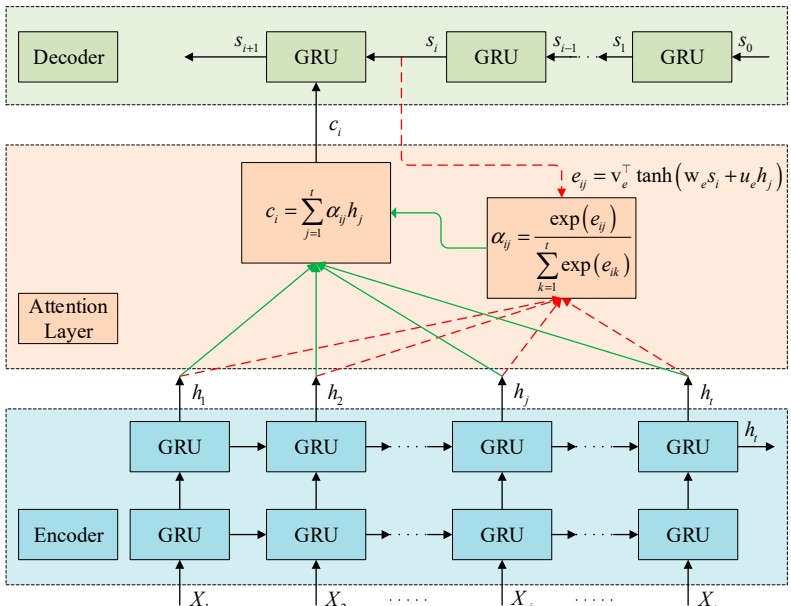

**Figure 2.** Temporal attention layer structure.

Assume that $X = [X_1, X_2, \ldots, X_t]^T$ is the input sequence of power load data. After the input data has passed through a multi-layer GRU network, the hidden state of each time step of the GRU network is used as the output of the encoder, instead of the hidden state output of only the last time step of the traditional encoder-decoder network. So, the encoder maps the input sequence $X$ to $h = [h_1, h_2, \cdots, h_j, \cdots, h_t]^T$. In the decoder, the hidden state $s_i$ output from the step $i$ of the decoder is compared with the encoder output for attention weighting:

$$e_{ij} = v_e^\top \tanh(w_e s_i + u_e h_j) \tag{3}$$

where $e_{ij}$ denotes the similarity of the hidden state at the step $i$ to the output of the encoder at step $j$. $v_e$, $w_e$ and $u_e$ are the parameters to be learned and the dimension of $v_e$ is 1, the dimensions of $w_e$ and $u_e$ are the same hyperparameters to be optimized. With obtaining the similarity of $s_i$ the encoder output at each step, the percentage of each similarity in the whole is calculated:

$$\alpha_{ij} = \frac{\exp(e_{ij})}{\sum\limits_{k=1}^{t} \exp(e_{ik})} \tag{4}$$

where $\alpha_{ij}$ denotes the weight of the hidden state at step $i$ concerning the output of the encoder at step $j$. This Equation is calculated to ensure that all attention weights sum to 1. After the attention weights are obtained, the encoder output is weighted and summed, and the encoding vector $c_i$ is calculated as follows:

$$c_i = \sum\limits_{j=1}^{t} \alpha_{ij} h_j \tag{5}$$

where the encoding vector $c_i$ will be decoded as the input of the $i + 1_{st}$ GRU unit. And the initial state $s_0$ of the decoder is the hidden state of the last time-step output of the encoder.

### 2.3. Attention-Based Codec Prediction Model

With the temporal attention layer, this paper proposes an attention-based codec prediction model, which mainly consists of three parts (as shown in Figure 3): An encoder composed of a multilayer GRU network, an attention layer, and a decoder composed on basis of multilayer GRU network. The encoder is responsible for encoding the electric load data, and the final hidden state of the encoder is used as the initial state of the

decoder. The attention layer is responsible for computing the attention weights between the encoder output and the decoder hidden states to obtain the encoding vector. The decoder is responsible for decoding the encoded vector and obtaining the predicted values.

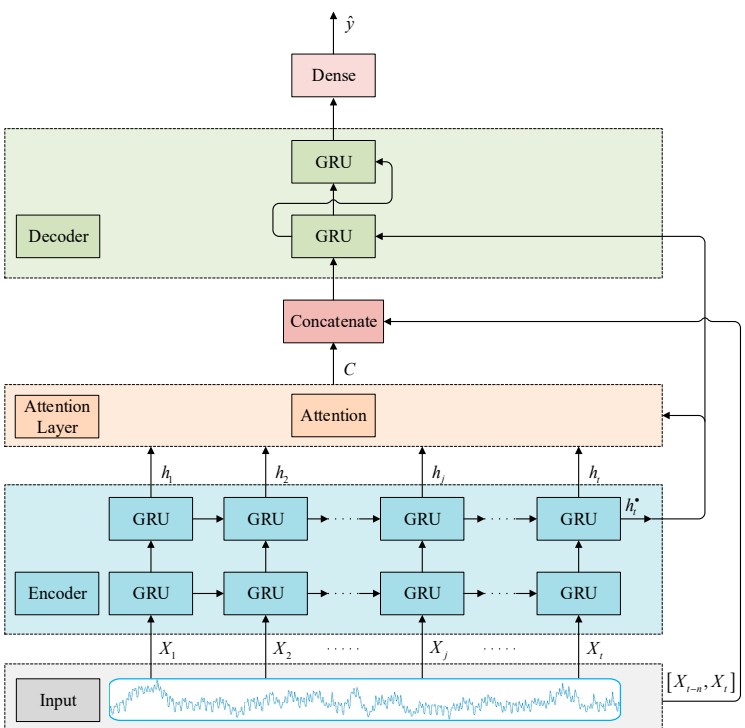

**Figure 3.** Structural description of attention-based codec prediction model.

As shown in Figure 3, the power load data is firstly sliding windowed to obtain the input data $X = [X_1, X_2, \ldots, X_t]^T$ and the true value $y = [y_{t+1}, y_{t+2}, \ldots, y_{t+m}]^T$. The encoder network consisting of a multilayer GRU network encodes the input $X$ to obtain the encoder output $h = [h_1, h_2, \cdots, h_j, \cdots, h_t]^T$ and the hidden unit state $h_t^\bullet$ of the last GRU layer of the encoder. This hidden state has two uses, one is as the initial state of the decoder layer GRU; The other is to compute attention weights with the encoder output. In the attention layer, the similarity between the hidden cell state $h_t^\bullet$ and the encoder output $h_j$ is first calculated separately, and then Softmax is used to ensure that the sum of the similarities is 1. Then the attention weights are weighted and summed with the encoder output to obtain the encoding vector $C$. The decoder input vector is obtained by splicing the coding vector $C$ with the last $[X_{t-n}, X_t]$ numbers of the input vector, and the decoder forward is calculated as:

$$inp\_de = [C, [X_{t-n}, X_t]] \tag{6}$$

$$z_t = \sigma(W_z[h_t', inp\_de] + b_z) \tag{7}$$

$$r_t = \sigma(W_r[h_t', inp\_de] + b_r) \tag{8}$$

$$\widetilde{h}_t = \tanh(W_h[h_t' \odot r_t, inp\_de] + b_h) \tag{9}$$

$$h_t = (1 - z_t) \odot \widetilde{h}_t + z_t \odot h_t' \tag{10}$$

The *decoder_output* is obtained after a last multilayer GRU network of decoder, and the decoder output is linearly varied to obtain the predicted value $\hat{y}$:

$$\hat{y} = W_o * decoder\_ouput + b_o \tag{11}$$

where $W_o$ and $b_o$ are the weight matrix and bias vector to be learned.

### 2.4. Bayesian Optimization for Global Hyperparameters

Before deep neural network training, we need to initialize the hyperparameters of the model to ensure the performance of the prediction model. However, the selection of network hyperparameters based on experience and a large number of attempts is not only time-consuming and computationally expensive for algorithm training but also does not always maximize the performance of the model [53]. Therefore, the screening process of the model's hyperparameters needs to be optimized to improve the robustness and accuracy of the whole model. In this paper, the Bayesian optimization method is applied to find and select the optimal evaluation points. The Bayesian optimization framework effectively uses the complete historical information to improve the search efficiency, and its most important theory is to constantly predict the posterior knowledge through the prior points [54]. Specifically, Bayesian optimization firstly assumes a functional relationship between the hyperparameters and the loss function to be optimized:

$$p^* = \underset{p \in P}{\text{argmin}}\, loss(p) \tag{12}$$

where $P$ is the set of all hyperparameters and $p$ is the set of hyperparameter combinations in $P$. $p^*$ is the optimal combination of parameters obtained from the final optimization, and $loss(\bullet)$ is the objective function needed to be optimized. In our model, the hyperparameters include attention layer similarity matrix dimensions of $w_e$ and $u_e$ in Equation (3), encoder network layers, decoder network layers, number of encoder network units, number of decoder network units, number of raw data input by the decoder, batch size of training data, number of training epochs, and model training optimizer. The loss function to be defined by the root mean square error (RMSE) as:

$$loss(p_j) = \sqrt{\frac{\sum_{i=1}^{n} (\hat{y}_i(p_j) - y_i)^2}{n}} \tag{13}$$

where $p_j$ is the $j$-th hyperparameter combination, $y$ is the true value, and $\hat{y}(p_j)$ is the model output results obtained using the $j$-th hyperparameter combination $p_j$.

The next process of Bayesian optimization is to construct the data set $D = [(x_1, y_1), (x_2, y_2), \cdots, (x_i, y_i), \cdots]$, where, $x_i$ is the $i$-th set of hyperparameters and $y_i$ is the error of the model output result under that set of hyperparameters:

$$y_i = loss(p_i) \tag{14}$$

The posterior probability $P(y|x, D)$ is derived from the data set $D$. The alternative model M follows a Gaussian distribution G with mean $\mu$ and variance $K$. And the specific functional expression $M$ is obtained by fitting the data set $D$:

$$p(loss) = G(loss; \mu, K) \tag{15}$$

$$p(loss\,|\,D) = G(loss; \mu_{loss|D}, K_{loss|D}) \tag{16}$$

Based on the resulting model $M$, the next observation is selected using the acquisition function $a(p)$:

$$p* = \text{argmax}\, a(P, p(y|x)) \tag{17}$$

In Bayesian decision theory, the capture function can be interpreted as evaluating the expected loss associated with $loss$ on the hyperparameter space $p$. After obtaining the parameters, the error of the model's output under the parameters is calculated. The parameters and loss are then updated to the data set $D$. The Bayesian optimization method builds a model based on historical data, evaluates the performance of the hyperparameters, and then selects new hyperparameters to test based on the model. The process is repeated continuously to obtain optimal parameters [55].

## 3. Results

### 3.1. Datasets and Setup

In this study, the electric power load data is from American Electric Power Company (AEP), which includes 26,280 data from 1 January 2017–1 January 2020, in which a sampling frequency is 1 h. The data set was divided into a training set and a test set in the ratio of 7:3 and normalized. The model is trained using a supervised learning approach, using window sliding, to divide the electricity load data into multiple sets of input and target values. Each set of data after the sliding window is arranged in a chronological pattern. This pattern may be learned by the neural network. To improve the generalizability of the model, each set of data after the sliding window is shuffle and then fed into the neural network for training. After determining the hyperparameters by the Bayesian optimization algorithm, the model is trained. After determining the hyperparameters by the Bayesian optimization algorithm, using the optimal hyperparameters training the model. During testing, the test data is fed into the trained model to obtain the prediction results. In this experiment, 24 electrical load data from the day $i$ are used as input objects and 24 electrical load data from the day $i + 1$ are used as expectations. Using 24 electrical load data for one day to forecast 24 electrical load data for a future day is more significant. The overall system flow diagram is shown in Figure 4.

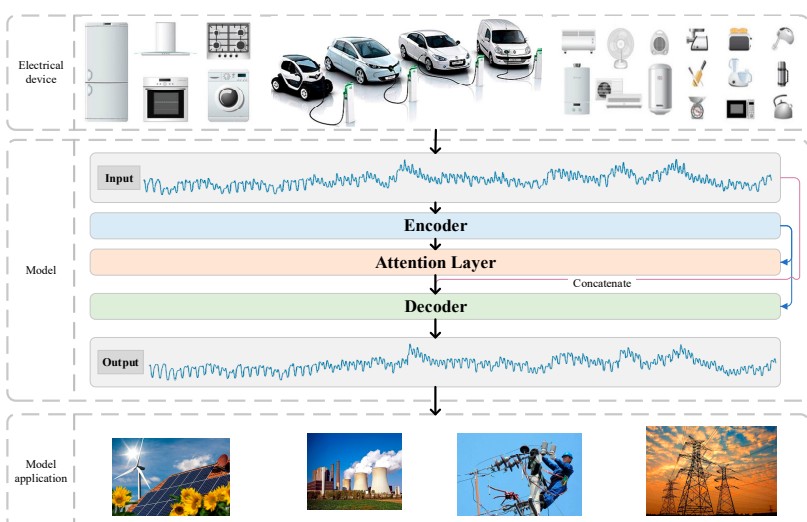

**Figure 4.** Load prediction system.

### 3.2. Evaluation Metrics

In this study, we used five indexes to evaluate the performance of the model, including root mean square error (RMSE), mean absolute error (MAE), Pearson correlation coefficient (R), normalized mean square error (NRMSE), and symmetric mean absolute percentage error (SMAPE).

The root means square error (RMSE) is the sum of the squares of the distances between the predicted and true values and measures the deviation of the observed value from the true value and reflecting the degree of dispersion of the predicted value. The smaller the value, the less the predicted value deviates from the true value. The mean absolute error (MAE) is used to measure the distance between the predicted and true values, which avoids the problem of errors neutralizing each other out and accurately reflects the magnitude of the actual prediction error. The Pearson correlation coefficient (R) is used to measure the linear correlation between the true and predicted values and varies from −1 to 1. It reflects the model's ability to fit non-linearly, and the larger the Pearson correlation coefficient, the better the fit between the predicted and true values. The normalized mean squared error is a transformation of the expression for the root mean squared error that allows an evaluation of the degree of variation in the data. The smaller the value the less the degree

of variation in the data and the more accurate the model prediction. The symmetric mean absolute percentage error is used to measure the proportion of deviation of the predicted value from the true value. The formulae for calculating the five indicators are as follows:

$$\text{RMSE} = \sqrt{\frac{\sum_{i=1}^{n}(\hat{y}_i - y_i)^2}{n}} \tag{18}$$

$$\text{MAE} = \frac{1}{n}\sum_{i=1}^{n}|\hat{y}_i - y_i| \tag{19}$$

$$R = \frac{\sum_{i=1}^{n}(\hat{y}_i - \overline{\hat{y}}_i)(y_i - \overline{y}_i)}{\sqrt{\sum_{i=1}^{n}(\hat{y}_i - \overline{\hat{y}}_i)^2 \sum_{i=1}^{n}(y_i - \overline{y}_i)^2}} \tag{20}$$

$$\text{NRMSE} = \frac{1}{\max(\hat{y}) - \min(\hat{y})}\sqrt{\frac{\sum_{i=1}^{n}(\hat{y}_i - y_i)^2}{n}} \tag{21}$$

$$\text{SMAPE} = \frac{1}{n}\sum_{i=1}^{n}\frac{|\hat{y}_i - y_i|}{(|\hat{y}_i| + |y_i|)/2} \tag{22}$$

where $n$ is the number of samples, $\hat{y}$ is the predicted value, $\overline{\hat{y}}$ is the average of the prediction, $y$ is the ground-truth value of the power load, $\overline{y}$ is the average of the ground-truth value.

The encoder-decoder model based on attention mechanisms has nine hyper-parameters, which are attention layer similarity matrix dimensions, encoder network layers, decoder network layers, number of encoder network units, number of decoder network units, number of raw data input by the decoder, batch size of training data, number of training epochs, model training optimizer. The Bayesian optimization algorithm was used to optimize the parameters of the model, which was constructed using GRU and LSTM, respectively. Table 1 shows the hyperparameters space of the Bayesian optimization algorithm. Where PL and PG are the encoder-decoder model of attentional mechanisms constructed with LSTM, and GRU, respectively:

**Table 1.** Bayesian optimization hyperparameter space and search results.

| Hyperparameter | Data Type | Values Range | PL | PG |
|---|---|---|---|---|
| Attention layer similarity matrix dimensions | integer | {1} + {2–64} step 2 | 18 | 4 |
| Encoder network layers | integer | {1–6} step 1 | 2 | 6 |
| Decoder network layers | integer | {1–6} step 1 | 5 | 2 |
| Number of encoder network units | integer | {8–128} step 8 | 120 | 72 |
| Number of decoder network units | integer | {8–128} step 8 | 120 | 72 |
| Number of data inputing decoder | integer | {0–24} step 1 | 6 | 23 |
| Batch size of training data | integer | {1} + {2–64} step 2 | 8 | 10 |
| Number of training epochs | integer | {60–200} step 5 | 130 | 155 |
| Model training optimizer | categorical | {Adam, Nadam, SGD} | Adam | Adam |

### 3.3. Comparative Prediction Results

In order to verify that our method could be used in the AEP dataset for predicting the electric load, we establish some experiments to compare the performance of our proposed models with that of other state-of-the-art deep-learning architectures. All models are trained and tested on a cloud server platform with Ubuntu 16.04 system. And codes are based on the open-source framework, Tensorflow2.3 with Python API, and run on a dual-core Intel Core i7-6800@3.6 GHz processor with two NVIDIA Tesla p40 GPUs, which have 48 G computing caches and 256G memory. Some classical RNNs are given in this section to illustrate the nonlinear and dynamic nature of load prediction problems, which has achieved remarkable success in other fields. Then nine methods, including Dense, RNN, LSTM, GRU, LstmSeq, GruSeq, LstmSeqAtt, GruSeqAtt, BLstmSeqAtt, are applied to further explain the effectiveness of temporal attention and Bayesian optimization in our

method for encoder-decoder prediction. The training is proceeded on the training set (70% proportion of AEP datasets), after that the evaluation is performed on the validation set (10% proportion) for minimizing overfitting. When the training process and parameter selection are achieved, the final evaluation is done on the unknown testing set (20% proportion) for evaluating the performance. All models use the Adaptive Moment Estimation (Adam) optimization algorithm, which uses momentum and adaptive learning rates to speed up convergence, and it is computationally efficient and has a low memory footprint. The loss function for model training is the mean absolute error (MSE).

$$\text{MSE} = \frac{\sum_{i=1}^{n} (\hat{y}_i - y_i)^2}{n} \tag{23}$$

where $n$ is the number of samples, $\hat{y}$ is the predicted value, $y$ is the ground-truth value of the power load. The MSE is derivable everywhere, the gradient values are dynamically changing and can converge quickly.

In detail, we set the number of network layers for Dense, RNN, LSTM, and GRU to 4 layers and the number of cells per layer to 24, and set the number of encoder layers of Lstmseq, Gruseq, LstmSeqAtt, and GruSeqAtt as 2, the number of decoder layers as 2, and the number of cells in each layer as 24. To fully verify the effectiveness of the proposed method, we repeated each method independently 20 times to ensure the objectivity of the results. The statistical results are shown in Figures 5 and 6.

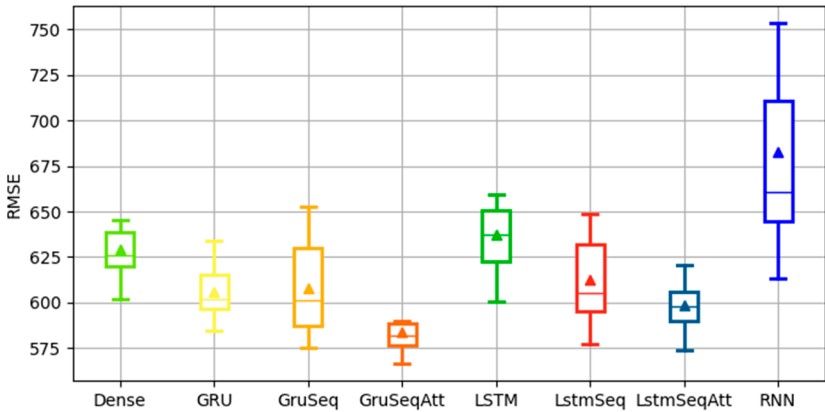

**Figure 5.** RMSE box line diagram for different models.

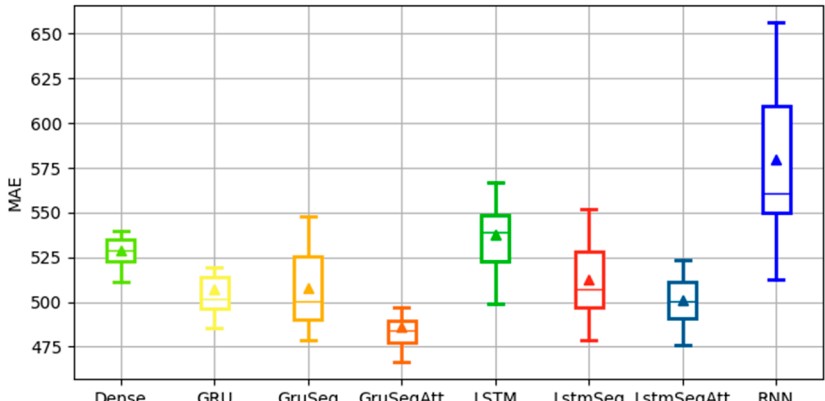

**Figure 6.** MAE box line diagram for different models.

It can be seen from Figures 5 and 6 that the distribution of RNN model results is the most discrete and the stability of the model is the worst. Lstmseq model is not as stable as the LSTM model, but the prediction error is smaller than the LSTM model. After integrating the attention mechanism, the stability of the model is better than before. The

GruSeq model is less stable and less accurate than the GRU network, probably because the encoder only uses the hidden state of the last time step as the encoded output, leading to missing encoded information as the input data gets longer and the later feature information masks the previous useful feature information. This is also the bottleneck of the sequence-to-sequence model. The encoder-decoder model incorporating the attention mechanism weights the information at each time step, thus overcoming the problem of information loss. As can be seen from Figures 5 and 6, the proposed method has the smallest box, the most concentrated distribution of RMSE and MAE, and the smallest average model error. Maintains a high level of accuracy and stability compared to other models.

The predictions for the test set using the proposed method and the comparison method are shown in Figure 7. The figure showing the curve of forecast results from 5 May 2019–11 May 2019. The partial zoom section shows the electricity load forecast curve for 7 May 2019. Due to the volatile and random nature of electricity load data, it is impossible for models to accurately predict the value of electricity load at each point in time. As can be seen from the partially enlarged part of Figure 7, the prediction results of our proposed method are closest to the true values.

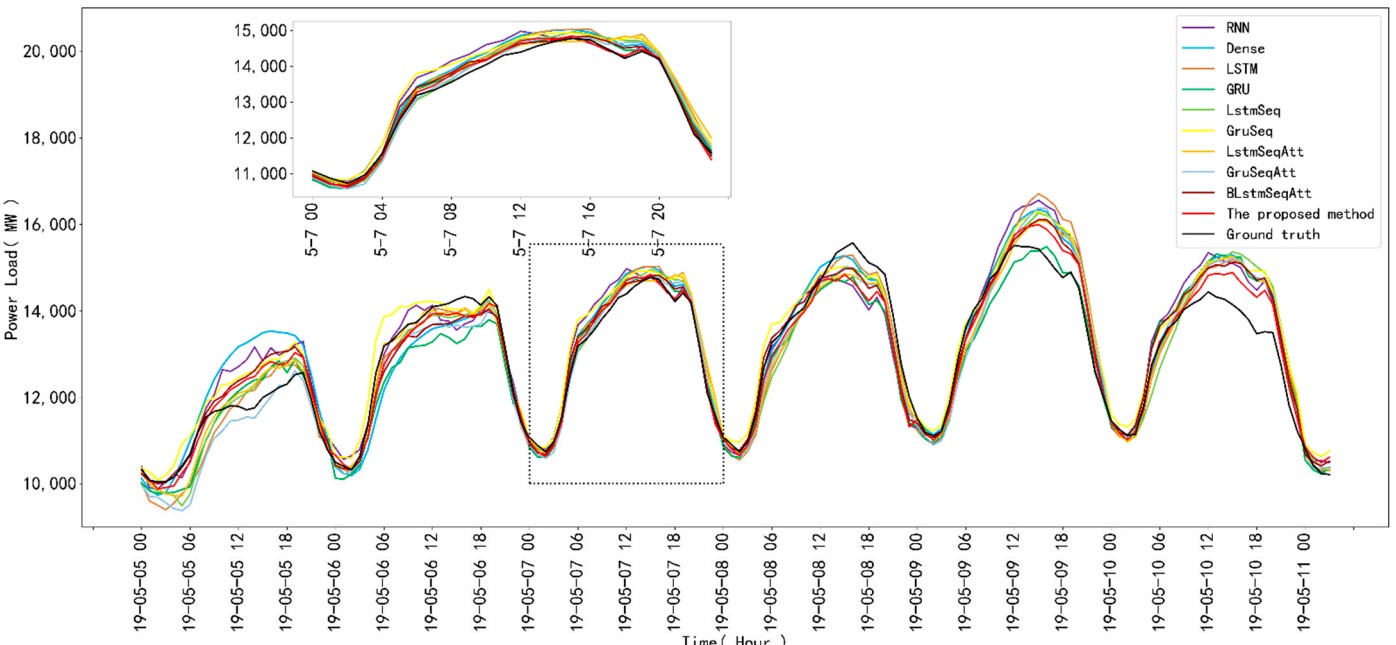

**Figure 7.** Prediction results of the power load by different models.

For short-term electricity load forecasting, it is of practical importance to predict the next day's peak load. we selected four typical days for further study. The 4 days are the Spring Equinox, Easter, Halloween, and Christmas. As can be seen from Figure 8, The proposed model can predict the trend of the next day's power load very well and can predict the next day's peak load. There were two load peaks on Spring Equinox and Christmas and one load peak on Easter and Halloween. There were two load peaks on the day of the Spring Equinox, at 07:00 and 19:00, with the maximum load peak of 17,194.32 MW on that day. There is a peak load at 12:00 on Easter Day with a peak load of 14,406.14 MW. There is a peak load at 18:00 on Halloween day with a peak load of 12,927.21 MW. There were two load peaks on Christmas Day, at 09:00 and 18:00, with a maximum load peak of 15599.39 MW on the day.

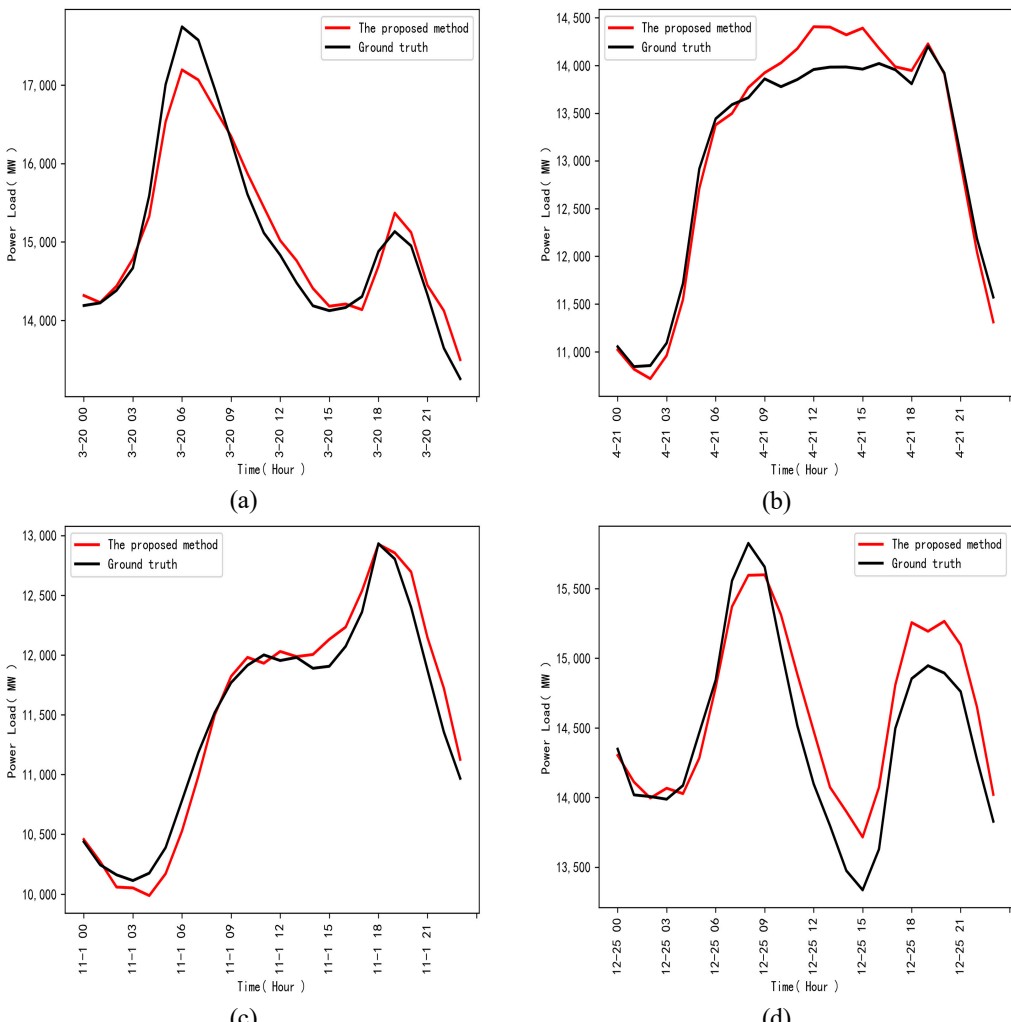

**Figure 8.** Prediction results for (**a**) 20 March 2019, (**b**) 21 April 2019, (**c**) 7 November 2019, and (**d**) 25 December 2019.

Figures 9 and 10 show the comparison results of the five indicators, respectively. It is clear from the graph that the proposed method has the smallest RMSE, MAE, NRMSE, and SMAPE, the largest R, the smallest error from the true value, and the highest degree of fit. The RMSE and MAE of LSTM, LstmSeq, LstmSeqAtt, and BLstmSeqAtt decreased in order when the model was constructed using LSTM as the basic unit. The RMSE and MAE of GRU, GruSeq, GruSeqAtt, and BGruSeqAtt were similarly reduced sequentially when the model was constructed using GRU as the base unit. It can be seen that the use of the encoder-decoder model structure has improved the performance of the model, and the incorporation of the attention mechanism has further improved the performance of the model. The model has the best performance after obtaining the optimal hyperparameters of the model using a Bayesian optimization algorithm. Moreover, comparing LSTM and GRU, LstmSeq and GruSeq, LstmSeqAtt and GruSeqAtt, and BLstmSeqAtt and BGruSeqAtt models, models using GRU units have better predictive performance than those using LSTM units.

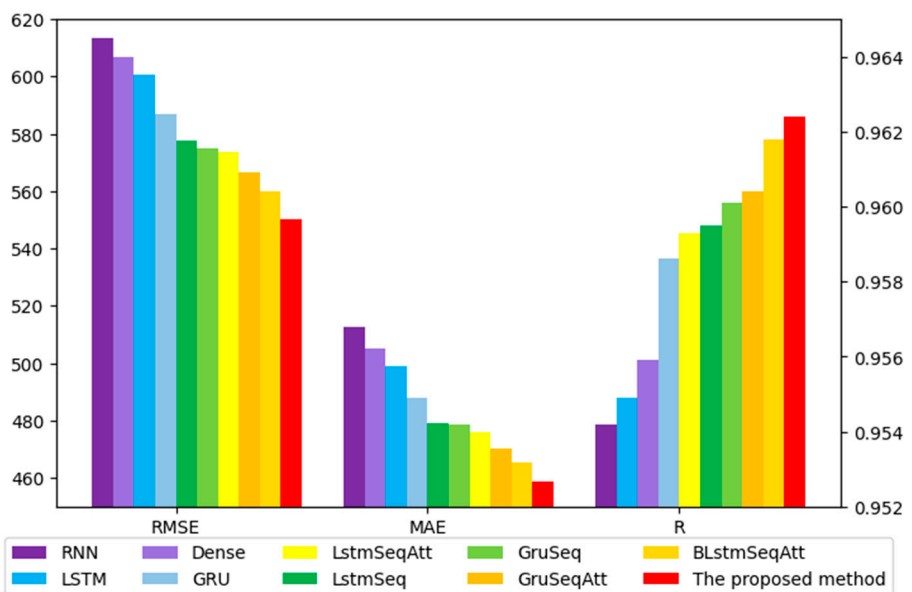

**Figure 9.** RMSE, MAE, and R of different models.

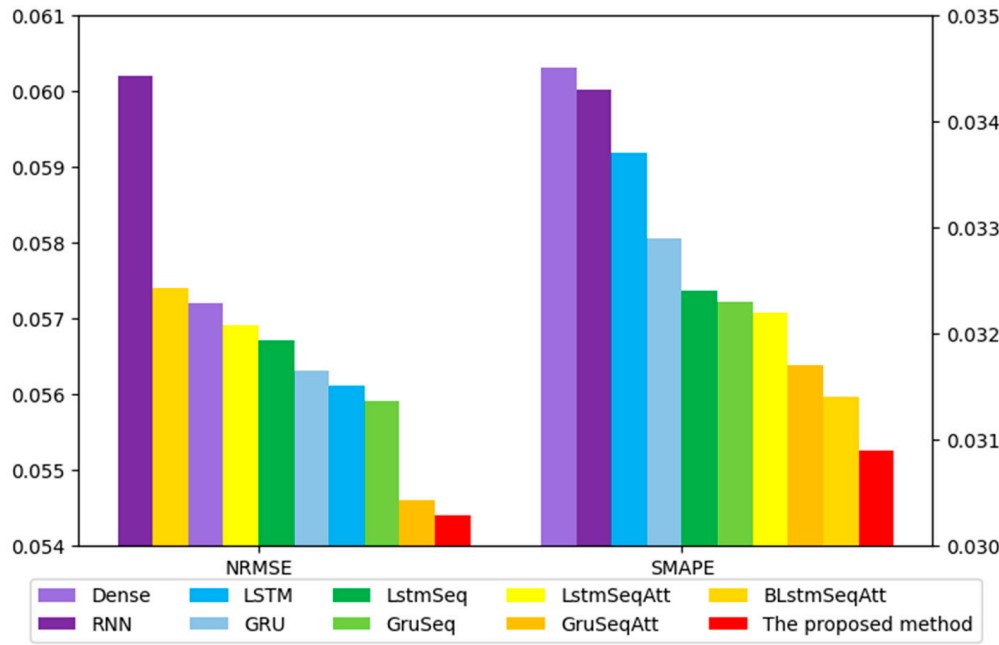

**Figure 10.** NRMSE and SMAPE of different models.

Table 2 shows the errors in the prediction results of the different models. We can see that the RMSE, MAE, R, SMAPE, NRMSE of the proposed is 550.3955, 458.9382, 0.9624, 0.0309, 0.0544, respectively, and the best prediction and all indicators are optimal. BGruSeqAtts reduce the RMSE 9.3% of Dense, 10.2% of RNN, 8.3% of LSTM, 6.2% of GRU, 4.7% of LstmSeq, 4.3% of GruSeq, 4.1% of LstmSeqAtt, 2.8% of GruSeqAtt, 1.7% of BLstmSeqAtt. Similarly. Similarly, BGruSeqAtts reduce the MAE 9.1% of Dense, 10.5% of RNN, 8.0% of LSTM, 6.0% of GRU, 4.2% of LstmSeq, 4.1% of GruSeq, 3.6% of LstmSeqAtt, 2.4% of GruSeqAtt, 1.4% of BLstmSeqAtt. Our proposed method has the smallest prediction error, the best fit to the true value, and the smallest deviation from the true value.

**Table 2.** Errors of the prediction result in different models.

| Model | RMSE | MAE | R | SMAPE | NRMSE |
|---|---|---|---|---|---|
| Dense [56] | 606.7183 | 504.9654 | 0.9559 | 0.0343 | 0.0602 |
| RNN [57] | 613.2839 | 512.6149 | 0.9542 | 0.0345 | 0.0572 |
| LSTM [58] | 600.4019 | 498.8333 | 0.9549 | 0.0337 | 0.0561 |
| GRU [59] | 586.8837 | 487.9156 | 0.9586 | 0.0329 | 0.0563 |
| LstmSeq [60] | 577.462 | 479.0576 | 0.9595 | 0.0324 | 0.0567 |
| GruSeq | 575.1462 | 478.5666 | 0.9601 | 0.0323 | 0.0559 |
| LstmSeqAtt | 573.7516 | 475.9956 | 0.9593 | 0.0322 | 0.0569 |
| GruSeqAtt | 566.5466 | 470.2515 | 0.9604 | 0.0317 | 0.0546 |
| BLstmSeqAtt | 560.0931 | 465.4894 | 0.9618 | 0.0314 | 0.0574 |
| Proposed method | 550.3955 | 458.9382 | 0.9624 | 0.0309 | 0.0544 |

## 4. Conclusions

Short-term electrical load forecasting plays an important role in the safety, stability, and sustainability of the power production and scheduling process. Better prediction results can help the electricity industries and power supply companies make reliable decisions to control the operation status, manage power systems and facilitate, reducing costs, and prevent pollution. In this paper, an attention-based encoder-decoder network with Bayesian optimization was proposed to forecast short-term power load, which gives full play to the GRU's powerful feature extraction and learning capabilities encoder-decoder neural network for time series data modeling. The temporal attention layer focusing on the key features of input data promotes the forecasting model's prediction accuracy and robustness. Finally, the Bayesian optimization method is used to confirm the model's hyperparameters for achieving the optimal predictions. The verification experiments with short-term electricity load data from American Electric Power (AEP) datasets show that the proposed method has the best performance in terms of prediction accuracy and stability compared to other models, which takes various indicators, RMSE, MAE, R, SMAPE, NRMSE, reflecting accuracy performance, response speed, and computing consumption into account.

We will attempt to examine our proposed model on a more complex time series dataset in future research. Moreover, we would like to apply advanced deep-learning network and optimization algorithms to adjust hyperparameter searching and model training to improve the prediction method's overall performance even further. Lastly, the model proposed in this study can be applied to power prediction and other field applications that contain multiple temporal information.

**Author Contributions:** Conceptualization, X.-B.J. and W.-Z.Z.; data curation, S.L.; formal analysis, X.-B.J. and Y.-T.B.; methodology, J.-L.K. and W.-Z.Z.; software, W.-Z.Z.; supervision, J.-L.K.; validation, J.-L.K. and X.-Y.W.; visualization, W.-Z.Z. and T.-L.S.; writing—original draft, X.-B.J. and X.-Y.W.; writing—review and editing, J.-L.K. and S.L.; All authors have read and agreed to the published version of the manuscript.

**Funding:** This work was supported in part by the National Natural Science Foundation of China (No. 62006008, 61903009), National Key Research and Development Program of China (No. 2017YFC1600605); Beijing Municipal Education Commission (No. KM201910011010, KM201810011005), Beijing excellent talent training support project for young top-notch team (No. 2018000026833TD01) and Humanities & Social Sciences of Ministry of Education (No. 19YJC790028).

**Institutional Review Board Statement:** Not applicable.

**Informed Consent Statement:** Not applicable.

**Data Availability Statement:** Data used to support the findings of this study are available from the corresponding author upon request.

**Conflicts of Interest:** The authors declare no conflict of interest.

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
