# Peer review of "Deep-Learning Forecasting Method for Electric Power Load via Attention-Based Encoder-Decoder with Bayesian Optimization"

_energies, doi:10.3390/en14061596_

Round 1

Reviewer 1 Report

Background section:

Terminology:

  • Deep learning is a machine learning subfield. I am aware that some reviews seem to draw a distinction on this line but the terminology is still wrong (l67-69)
  • There are approaches of converging ANN to useful weights other than backpropagation, however, the term backpropagation neural network is not especially established in the literature as a term distinct from artificial neural network. I would recommend staying within established terminology and citing both sources as ann (l84-85)
  • Albeit deep belief networks are a slightly different category, a ‘deep neural network’ is as arbitrary as it gets and a hyper-category of deep CNN, deep encoder-decoder models, deep belief networks etc. This characterization of sources seems along an arbitrary line (l109-111)

General:

  • There is sufficient literature to show that most (if not all) mentioned training tricks can also be successfully applied to ‘non-dl’ ml methods. (l103-107)
  • Please also cite the original LSTM paper which first proposed this neuron/cell type and already provided strong empirical evidence for LSTM’s better time-series modeling capacity (https://dl.acm.org/doi/10.1162/neco.1997.9.8.1735). Citing source 35 for further, application specific evidence in this context is obviously acceptable, but please avoid citing in a way that implies that 35 proposed the LSTM! (l119-122)
  • Section l134-143 is duplicated…
  • Going with the above-trend (attention to detail and terminology), please cite the original s2s paper (https://arxiv.org/abs/1409.3215). Additionally, encoder-decoder models are a different architecture type and classical monolithic architectures. This means that they are in principle independent of the cell-type used. RNN, as well as feedforward type cells/neurons, can and have successfully been used in such models (see for example https://arxiv.org/abs/1706.03762). (l144-148) you even carry on in the next sentence with encoder-decoder models consisting of several RNN layers.
  • ‘nevertheless, the encoder-decoder structure has not yet been well researched for pre- diction modeling of electric load.’ (l160f) this is simply not true. There are a plethora of works attempting to apply s2s-type architectures to load forecasting (https://scholar.google.ca/scholar?q=forecasting+load+sequence+to+sequence&hl=en&as_sdt=0,5&as_vis=1). This of course does not imply that there is no need for further research into how to apply encoder-decoder structures to multivariate time-series forecasting tasks such as load forecasting. But the lack of rigorous analysis of closely related, existing work is troublesome!
  • Bayesian optimization of machine learning algorithms using Gaussian processes (https://arxiv.org/abs/1206.2944) should also be cited. (l171) using such approaches to optimize complex neural networks is an established practice and I have difficulties agreeing with the author’s claim of this as a novel contribution to the field.

In short, the background section fails to establish a clear terminology as well as an accurate frame of reference based on which to judge the quality of the model proposed in further sections.

Model proposal section:

  • Section 2.2
    • The information bottleneck within attention-less s2s systems has been documented in earlier sources by the groups that proposed using termporal attention with such models in the first place (https://arxiv.org/abs/1409.0473, https://arxiv.org/abs/1508.04025). This section mis-attributes specific contributions (l224-229).
    • Citing source 46 as additional source within the context of time series forecasting is of-course acceptable, but this is not the original proposing work!
  • Section 2.4:
    • As mentioned above, this section is missing a citation for https://arxiv.org/abs/1206.2944 or other similarly early proposed methods.

Experiment section:

  • As expected, s2s models with attention perform better than s2s models without attention.
  • As expected, s2s models tend to perform better than monolithic architectures
  • Interestingly enough there seems to be a performance difference between GRU and LSTM based models. I would like the experiment section to also include the number of parameters that each model has. One LSTM cell has more learnable parameters than a GRU cell, which means comparing on a ‘number of neurons/layers’ basis is slightly flawed.
  • Since the proposed method is basically adapting an attentive s2s model based on GRU’s and then performing heavy hyperparameter fine-tuning I would like the experiments to also include some of the other models with fine-tuned parameters. As of now, the experiments only compare untuned alternatives. It seems a logical experiment to tests the validity of both components of the paper’s proposed model by also including hyperparameter-tuned variants of the other models. I expect the ‘fine-tuned’ performance of the LSTM based models and of the GRU based models to be very close to each other. At the moment it is obvious that the fine-tuned model will outperform the not fine-tuned models.

Copied from comments for editors for transparency:

The authors present a sequence to sequence model with attention based on gated recurrent units to forecast short-term electrical load demand and fine-tune the model hyperparameters using Gaussian processes.

Some of the backgrounds is slightly miss-cited or miss-attributed, the terminology used at times inaccurate and the proposed process does not offer any novelties on the machine learning side.

Additionally, similar, already existing work is not cited.

The application presented (S2S models for load forecasting, or in general the wider application of more modern deep learning models) is nonetheless understudied and the model does outperform several other published sources.

The experiments aim to offer some comparison but do so on a skewed playing field. The model used is built on two principal components (architecture and parameter tuning), but the experiments only compare non fine-tuned alternate architectures. The experiments should be extended to also include fine-tuned versions of alternate architectures to provide a true comparison.

Additionally, the paper exhibits some grammar mistakes.

As such, this article should be accepted only after major revision, addressing above mentioned issues.

As it is, the article offers little new insight (other works proposing essentially the same model on essentially the same application do exist, in contrast to the author’s claims) and does not elevate itself over other similar works by either providing more extensive experiments or more in-depth analysis.

Reviewer 2 Report

  • The forecast horizon is the short term. How does the proposed  method is different from ANN regressive methods ?
  • The exogenous variables have been studied?
  • The short-term forecast has relevant information. But the most significant thing is to predict the next day's load Peak. How the model behaves?

The authors should see these papers.

  • Santos P. J., Martins A, Pires. 2007. Designing the input vector to ANN-based models for short-term load forecast in electricity distribution systems International Journal of Electrical Power & Energy System: 29: 338-347, ELSEVIER DOI: 10.1016/j.ijepes. 2006.09.002
  • S. Chemetova; P. J. Santos; M. Ventim-Neves, 2016 Load peak forecasting in different load patterns situations 10th International Conference on Compatibility, Power Electronics and Power Engineering (CPE-POWERENG) Pages: 148 151 DOI: 10.1109 /CPE.2016.7544175

  •  

Reviewer 3 Report

This paper presents an encoder-decoder architecture (based on sequence-to-sequence learning) for the 24-ahead deterministic prediction of electric load. The model is complemented with an attention mechanism to optimally capture time dependencies, as well as an Bayesian optimization algorithm for helping the cumbersome task of finding the optimal hyper-parameters. The methodology is interesting, well explained and well justified. The case study is convincing. Overall, the work is valuable, but here are some suggestions to further improve the paper.

The abstract should be more specific on the targeted time horizon (i.e., 24 hours ahead) and on the fact that the paper focuses on deterministic (point) forecasting, thus neglecting the uncertainty regarding the predictions. This is important to define the scope of the paper for the interested readers.

Also, in page 1, lines 15-16, one may kindly disagree with the following sentence "Due to the limitations of existing load predicting methods in dealing with massive, nonlinear, and dynamic time-series data ». This is not the first work that is tailored to tackle such effects, and the sentence should be rewritten to be more realistic.   In the literature review, a differentiation is made between machine learning and deep learning, but one may argue that deep learning is a subspace of machine learning. I am not sure that this distinction is fully correct, and it may be worthwhile to revise the structure of the subsequent paragraph.   The literature review should also be completed with works that closely relates to the proposed paper. In this way, the encoder-decoder architecture using GRU (without attention mechanism) has already been introduced in the power systems literature, i.e.,  [] J. Bottieau, L. Hubert, Z. De Grève, F. Vallée and J. Toubeau, "Very-Short-Term Probabilistic Forecasting for a Risk-Aware Participation in the Single Price Imbalance Settlement," in IEEE Transactions on Power Systems, vol. 35, no. 2, pp. 1218-1230, March 2020.  [] Hyper-parameter Optimization of Multi-attention Recurrent Neural Network for Battery State-of-Charge Forecasting.   Although it can be justified for some application to rely on deterministic forecast (and the paper has thus a strong practical interest), the introduction should also discuss the possibility to extend the work to probabilistic predictions (to quantify the inherent uncertainty around predictions), such as done in, e.g., [] Y. Wang, D. Gan, M. Sun, N. Zhang, Z. Lu, C. Kang, "Probabilistic individual load forecasting using pinball loss guided LSTM," Applied Energy, vol. 235, pp. 10-20, 2019. [] J. Toubeau, J. Bottieau, F. Vallée and Z. De Grève, "Deep Learning-Based Multivariate Probabilistic Forecasting for Short-Term Scheduling in Power Markets," in IEEE Transactions on Power Systems, vol. 34, no. 2, pp. 1203-1215, March 2019.   Overall, it should be noted that both encoder and decoder architecture can be built with any type of architecture (not necessarily recurrent networks as mentioned in page 4, line 200).   Regarding the selection of hyperparameters, it should also be noted that, depending on the choice of the optimization algorithm, additional hyperparameters (such as learning rate) may have to be added to the list.   It seems that the model is only fed with historical information (there are no other inputs). Is it correct? Please be more specific in the revised version of the paper.   The units are missing in many figures. Also, Figure 7 has no labels nor units (associated to those labels).

Round 2

Reviewer 1 Report

The authors addressed all my concerns from the first round of reviews. One item that slipped my attention during the previous round is that of the experiment reproducibility: I cannot seem to find a description of what algorithm and loss function is used to train the NN. I would expect this kind of information to be included in the paragraphs from lines 407-429. This is an important detail but easy to remedy.

There are still some further language issues to fix, e.g. on lines 68-70, 70-73, and 231-237.
